# Glucose, Fructose, and Urate Transporters in the Choroid Plexus Epithelium

**DOI:** 10.3390/ijms21197230

**Published:** 2020-09-30

**Authors:** Yoichi Chiba, Ryuta Murakami, Koichi Matsumoto, Keiji Wakamatsu, Wakako Nonaka, Naoya Uemura, Ken Yanase, Masaki Kamada, Masaki Ueno

**Affiliations:** 1Department of Pathology and Host Defense, Faculty of Medicine, Kagawa University, 1750-1 Ikenobe, Miki-cho, Kita-gun, Kagawa 761-0793, Japan; ychiba@med.kagawa-u.ac.jp (Y.C.); ryuta@med.kagawa-u.ac.jp (R.M.); inpathma@med.kagawa-u.ac.jp (K.M.); s20d727@stu.kagawa-u.ac.jp (K.W.); 2Department of Supportive and Promotive Medicine of the Municipal Hospital, Faculty of Medicine, Kagawa University, 1750-1 Ikenobe, Miki-cho, Kita-gun, Kagawa 761-0793, Japan; w_nonaka@med.kagawa-u.ac.jp; 3Department of Gastroenterology and Neurology, Faculty of Medicine, Kagawa University, 1750-1 Ikenobe, Miki-cho, Kita-gun, Kagawa 761-0793, Japan; 4Department of Anesthesiology, Faculty of Medicine, Kagawa University, 1750-1 Ikenobe, Miki-cho, Kita-gun, Kagawa 761-0793, Japan; uemu@med.kagawa-u.ac.jp (N.U.); yanaken@med.kagawa-u.ac.jp (K.Y.); 5Department of Neurological Intractable Disease Research, Faculty of Medicine, Kagawa University, 1750-1 Ikenobe, Miki-cho, Kita-gun, Kagawa 761-0793, Japan; kamada93@med.kagawa-u.ac.jp

**Keywords:** blood-cerebrospinal fluid barrier, GLUT, SGLT, URAT1, BCRP/ABCG2, metabolism, renal proximal tubule, cell polarity, aging, neurodegenerative diseases

## Abstract

The choroid plexus plays a central role in the regulation of the microenvironment of the central nervous system by secreting the majority of the cerebrospinal fluid and controlling its composition, despite that it only represents approximately 1% of the total brain weight. In addition to a variety of transporter and channel proteins for solutes and water, the choroid plexus epithelial cells are equipped with glucose, fructose, and urate transporters that are used as energy sources or antioxidative neuroprotective substrates. This review focuses on the recent advances in the understanding of the transporters of the SLC2A and SLC5A families (GLUT1, SGLT2, GLUT5, GLUT8, and GLUT9), as well as on the urate-transporting URAT1 and BCRP/ABCG2, which are expressed in choroid plexus epithelial cells. The glucose, fructose, and urate transporters repertoire in the choroid plexus epithelium share similar features with the renal proximal tubular epithelium, although some of these transporters exhibit inversely polarized submembrane localization. Since choroid plexus epithelial cells have high energy demands for proper functioning, a decline in the expression and function of these transporters can contribute to the process of age-associated brain impairment and pathophysiology of neurodegenerative diseases.

## 1. Introduction

The composition of the interstitial fluid (ISF) of the central nervous system (CNS) is tightly controlled by the blood-brain barrier (BBB) and blood-cerebrospinal fluid barrier (BCSFB) in order to ensure the appropriate microenvironments for neuronal signaling [1,2]. The BBB is composed of endothelial cells interconnected by tight junctions, the basement membrane, pericytes, and the end feet of astrocytes, with high transendothelial electrical resistance ranging between 1500 and 2000 Ω∙cm^2^ [1,3,4]. Endothelial cells of capillaries in the choroid plexus (CP) parenchyma are fenestrated, with the pores measuring 30 to 50 nm in diameter, allowing a facilitated diffusion of molecules with molecular weights up to ~800 kDa [3,5,6,7]. In contrast, the entry of macromolecules into the cerebrospinal fluid (CSF) is restricted by the BCSFB, which is composed of a monolayer of CP epithelial cells joined by tight junctions. The electrical resistance of the CP epithelial cell layer is in the intermediate range (~150 Ω∙cm^2^), indicating its permeable nature similar to some kidney and gut segments [1,4,8,9]. These two structures function as the interface between blood and brain fluid (i.e., brain ISF and CSF), preventing plasma proteins such as albumin and circulating blood cells (erythrocytes and leukocytes) from entering freely into the brain parenchyma. Moreover, they protect the brain from physiological fluctuations in plasma components and from blood-borne neurotoxic substances [1,2,3]. In contrast to the BBB or BCSFB, the ependymal cells that line the ventricular wall to separate the CSF from the brain ISF are interconnected with gap junctions, making them permeable to most substances, including macromolecules [10].

In addition to their barrier function, the BBB and BCSFB are also responsible for maintaining the brain microenvironment by regulating the supply of essential nutrients such as glucose and amino acids, the exchange of electrolytes between the brain ISF and the circulating blood, and the removal of metabolic wastes and neurotoxic substances such as drug metabolites and amyloid β (Aβ) [1,2,3,11,12]. To fulfill such needs, brain capillary endothelial cells and CP epithelial cells are equipped with several specific transporter proteins [1,2,3]. However, these two cells show significant dissimilarities in the expression profiles of transporters and their localization, which ultimately leads to their functional differences [1,3,4]. One of the primary functions of the CP is to produce and secrete a major fraction of the CSF, about 500 mL a day in human adults [8,13]. To meet the requirements for such a high yield demand, the CP epithelial cells have a spectrum of ion and organic solute transporters that are distinguishable from those present in the BBB endothelial cells [1,10]. These transporting functions require high energy expenditure; therefore, the CP receives its blood supply at a high rate (4 mL/min per gram), which is 5–10 times higher than the blood supply rate to the brain parenchyma [10,13]. Moreover, the CP epithelial cells contain a large number of mitochondria, which comprise 12–15% of the adult cell volume [14].

This review focuses on the recent knowledge advances on the transporter proteins expressed in CP epithelial cells—in particular, those belonging to the facilitated diffusion glucose transporter (SLC2A/GLUT) and sodium/glucose cotransporter (SLC5A/SGLT) families, which are specific to hexose (glucose and fructose) and urate transport. These molecules are used by brain cells as either energy sources [1,15,16] or neuroprotective substances [17,18]. Herein is summarized the localization of these transporters in CP epithelial cells, discussing their physiological function, as well as the pathophysiological relevance of their expression at BCSFB in neurological diseases. It also describes the similarity of CP epithelial cells to renal proximal tubular epithelial cells, in terms of their morphological properties and the expression repertoire of transporter proteins.

## 2. Glucose Transporters in Choroid Plexus Epithelial Cells

Since glucose is a large polar molecule, it cannot cross the lipid bilayer of the plasma membrane by simple diffusion [19]. Therefore, cells are equipped with glucose transporters to enable glucose entry, which belong to either the SLC2A/GLUT [20] or SLC5A/SGLT [21] families. Except for GLUT1, which is ubiquitously expressed, each glucose transporter has a distinct cell- and tissue-specific expression pattern [16].

### 2.1. SLC2A Family

Proteins of the SLC2A family, encoded by *SLC2A* genes, are integral membrane proteins comprised of ~500 amino acid residues organized in 12 membrane-spanning domains with intracellularly located amino and carboxyl terminals and a single site of N-linked glycosylation [19,20]. The 14 members of the SLC2A/GLUT family are categorized into three classes based on sequence similarity: Class 1 (GLUTs 1–4 and 14); Class 2 (GLUTs 5, 7, 9, and 11); and Class 3 (GLUTs 6, 8, 10, 12, and HMIT) [20].

#### 2.1.1. GLUT1 (SLC2A1)

The brain is strictly dependent on glucose as its energetic source. Since the brain has almost no glucose storage, this macromolecule needs to be continuously supplied from the blood to brain cells to ensure normal brain functions [1,15,16]. GLUT1 plays a critical role on glucose transport from circulating blood to the brain parenchyma through the BBB [1,15,16,20]. Although GLUT1 is localized in both the luminal and abluminal membranes of the microvascular endothelial cells, it is asymmetrically distributed with four-fold higher expression on the abluminal membrane [16,22].

Similar to the BBB-forming endothelial cells, CP epithelial cells also use GLUT1 as the main glucose transporter [1,23]. Quantitative targeted absolute proteomics of human and rat CP revealed that GLUT1/Glut1 is the hexose transporter protein most abundantly expressed in CP [24]. A transcriptome study revealed that *Slc2a1*, which encodes Glut1 protein, is expressed in the rat CP at a three-fold higher level in E15 embryos than in adults [25]. Immunohistochemical studies further revealed that GLUT1/Glut1 is mainly present in the basolateral side of CP epithelial cell membranes (Figure 1a) [26,27]. In contrast to brain microvascular endothelial cells, the capillary endothelial cells in the CP parenchyma were negative for GLUT1/Glut1 (Figure 1a) [26].

#### 2.1.2. GLUT12 (SLC2A12)

Among other *Slc2a* family members that predominantly transport glucose, mRNAs for *Slc2a3*, *a4*, *a8*, *a12*, *a13*, and *a15* were identified in the rat CP; however, only *Slc2a12* is expressed at a functionally significant level in adult rats [25,29]. GLUT12/Glut12 is a member of the class 3 GLUT protein and is expressed in human and mouse brains at very low levels and in the mouse CP, as well as the heart, prostate, skeletal muscle, placenta, small intestine, and adipose tissue [20,30,31,32]. The role of GLUT12 in glucose homeostasis under physiological or pathological conditions remains to be clarified [20]. Nevertheless, some GLUT12 unique functional features were reported, including its capacity to increase glucose transport in the presence of extracellular sodium, as well as its potential to transport α-methyl-D-glucose, which is a substrate of the SGLT family [33]. Moreover, intriguingly, GLUT12/Glut12 expression was found to be upregulated in the frontal cortex of patients with Alzheimer’s disease (AD) [30] and in the brain of AD mouse models, aged mice, and in mice upon receiving intracerebroventricular injection of Aβ oligomers [34]. Although *Slc2a12* transcripts have been identified as being highly expressed in adult mice [35] and rats [25,29], the localization of the GLUT12 protein in the CP has not been confirmed by immunohistochemistry (Figure 1b) [28].

#### 2.1.3. GLUT3 (SLC2A3)

GLUT3 is the primary mediator of glucose uptake into neurons [15,20]. It is sometimes grouped together with GLUT14, which was identified as a duplicon of GLUT3 and exclusively expressed in human testes, due to their 94.5% identity in an amino acid sequence [36]. Transcriptome studies using rat CP suggested that *Slc2a3* was more expressed at the E15 developmental stage than in adults [25,29]. Immunohistochemical and immunoblot analyses did not confirm the expression of Glut3 protein in the CP [37], although a quantitative proteomic analysis of adult human CP revealed the expression of GLUT3 and 14 [24].

#### 2.1.4. GLUT4 (SLC2A4)

GLUT4 is mainly expressed in adipocytes, skeletal muscle, and cardiomyocytes, where it functions as an insulin-responsive glucose transporter [20,38]. In the brain, GLUT4 is present at the hippocampal nerve terminals and is mobilized by neuronal activity to support the energetic demands of firing synapses [39]. Several lines of evidence suggest that GLUT4 plays a pivotal role in hippocampal memory processing [40]. Transcriptomic and in-situ hybridization studies have revealed that *Slc2a4* is expressed only in embryonic rat and mouse CP [25,29,41].

### 2.2. SLC5A Family

Human *SLC5A* encodes 12 glycosylated membrane proteins of 60–80 kDa that have 14 membrane-spanning domains [21,42,43], four of which—SGLT1 (SLC5A1), SGLT2 (SLC5A2), SGLT4 (SLC5A9), and SGLT5 (SLC5A10))—are sodium/glucose cotransporters. SGLT3 (SLC5A4) is not a transporter but a glucosensor expressed in the enteric nervous system and muscle [44]. The other SLC5A family members cotransport sodium and myo-inositol (SMIT1 (SLC5A3) and SGLT6 (SMIT2 and SLC5A11)), iodide (NIS (SLC5A5)), multivitamin (SMVT (SLC5A6)), choline (CHT1 (SLC5A7)), and monocarboxylate (SMCT1 (SLC5A8) and SMCT2 (SLC5A12)) [43].

#### 2.2.1. SGLT1 (SLC5A1)

SGLT1 is a high-affinity and low-capacity glucose transporter that transports two sodium ions along with each glucose molecule [21,42,43]. SGLT1 is expressed in the intestinal epithelial cells [19,43,45,46] and in the apical membrane of the S3 segment of the renal proximal tubule [19,21,46,47,48]. Immunohistochemical studies revealed the presence of Sglt1 in neurons of the cerebral cortices, hippocampus, amygdala, hypothalamus, and cerebellar Purkinje cells in the rat brain [49,50]. It is still controversial whether SGLT1 is expressed in the BBB. The expression of Sglt1 was reported in rat and pig brain capillaries and in primary capillary endothelial cells from porcine or bovine brains [51,52], which was upregulated after transient ischemia [51] or oxygen-glucose deprivation [52]. In contrast, several studies failed to confirm a SGLT1/Sglt1 presence in immortalized or induced pluripotent stem cell-derived human brain microvascular endothelial cells and in the rat BBB by immunocytochemistry or immunohistochemistry [45,49,50,53]. Our immunohistochemical results also suggested the absence of SGLT1 in the capillaries of human brains (Figure 2a). In addition, the absence of transport across the BBB of α-methyl-4-deoxy-4-[^18^F]fluoro-D-glucopyranoside (Me-4-FDG), a SGLT-specific molecular imaging probe, suggests that SGLTs do not significantly contribute to the glucose transport into the brain [49,50,54].

Balen et al. reported the expression of Sglt1 in CP epithelial and ependymal cells in the rat brain, with CP epithelial cells showing granular immunoreactivity, suggesting localization in intracellular organelles [45]. Transcriptome studies of lateral ventricle CP of embryonic (E15) and adult (10-week-old) rats further reported that the expression of the *Slc5a1* transcript was 2.38-times higher in adults than in the E15 embryo [29,56]. However, reports on SGLT1 expression in the CP are conflicting [1], and we could not confirm the expression of SGLT1 in CP epithelial cells (Figure 2b) [55].

#### 2.2.2. SGLT2 (SLC5A2)

SGLT2 is a low-affinity and high-capacity glucose transporter with a stoichiometry of 1:1 for sodium and glucose [19,42,47,57]. It is predominantly localized at the brush border membrane of the proximal tubules (S1 and S2 segments) [21,42,47,58,59], where it is responsible for the uptake of 90% of the glucose from the glomerular filtrate [58]. SGLT2 inhibitors have been successfully used for the treatment of type 2 diabetes mellitus (T2DM) [46,60,61]. Although SGLT2 is mainly expressed in renal proximal tubular epithelial cells, studies have suggested its presence in other organs, including the cerebellum, hippocampus, isolated brain microvessels, heart, salivary gland, liver, thyroid gland, and pancreatic alpha cells [21,49,62,63]. Functional assays using rat midbrain slices indicated that Sglt2 is accountable for 20% of the total Me-4-FDG uptake [49]. In turn, Wright et al. suggested that SGLT2 (and SGLT1) expressed in the heart and brain may behave as glucose receptors [42,64], and the possibility that SGLT inhibitors may act in brain regions governing appetite and satiety to induce SGLT inhibitor-induced hyperphagia has not been conclusively ruled out [46]. SGLT2 in the brain may also affect the pathophysiology of epilepsy; however, the effects of SGLT2 inhibitors on murine epilepsy models are conflicting [65,66]. Brain SGLT2 and SGLT1 expressions are increased upon traumatic brain injury [67]. The expression of SGLT2 was also observed in tumor cells and proliferating microvasculature of high-grade astrocytomas; thus, Me-4-FDG could serve as a highly sensitive positron emission tomography (PET) probe for this malignant brain tumor [68]. In contrast to the tumor microvasculature, normal brain microvascular endothelial cells lack SGLT2/Sglt2 immunoreactivity (Figure 2c) [49,55].

We have recently reported the expression of SGLT2/Sglt2 in CP epithelial and ependymal cells in human and mouse brains [55]. Immunohistochemical staining revealed intracellular granular immunoreactivity in these CSF-facing cells with an anti-SGLT2 antibody raised against amino acid residues 220–266 of human SGLT2 (NBP1-92384, Novus Biologicals) (Figure 2d) [55], whereas immunohistochemistry using an antibody targeting the 250–350 residues of human SGLT2 (ab85626, Abcam) showed immunoreactivity on the basolateral membrane (Figure 2e, unpublished data). Furthermore, the expression of *Slc5a2* in mouse CP was confirmed by genetic analysis [55]. The functional significance of SGLT2 in CP epithelial cells remains to be elucidated.

### 2.3. Physiological and Pathophysiological Considerations of Glucose Transporters in Choroid Plexus Epithelial Cells

Due to the rapid consumption of glucose by brain cells, a downhill gradient of glucose from blood towards the brain ISF is maintained. Basal, steady-state brain ISF glucose levels in normoglycemic subjects (plasma glucose concentration of 6 mM) is estimated to be 1.4 mM [16], which matches well with the measured value in the rat brain ISF [69]. According to this concentration gradient, GLUT1 favors the blood-to-brain transport of circulating glucose.

The glucose level in the CSF is 50–80 mg/100 mL (2.78–4.44 mM) in normoglycemic individuals, or approximately 50–60% of the plasma glucose levels, which creates the downhill gradient from the blood to CSF, then to the brain ISF. The net glucose transfer from the blood to the CSF was confirmed using in-situ perfusion of sheep CP, and the process of glucose transport across CP was shown to be independent of the sodium ions [70]. The net glucose flux across the BCSFB is estimated at approximately 1.4 mmol/day, which is much less than the net flux of glucose across the BBB (about 600 mmol/day) [23]. The basolateral-predominant expression of glucose transporters (GLUT1 (Figure 1a), and maybe SGLT2 (Figure 2e)), in CP epithelial cells suggest that they may be important contributors for supplying glucose to CP epithelial cells in order to meet their high metabolic demands or for increasing the water permeability of the basolateral membrane [1,23]. The transport of water along with the solute has been established in SGLT1, as well as in several other symporter-type cotransporters [13,71,72]. In contrast to the abundant expression of glucose transporters (mainly GLUT1) in the basolateral membrane, information is scarce on the glucose transporters expressed on the apical membrane of CP epithelial cells. The expression of GLUT1 in the apical membrane of CP epithelial cells has been reported only in the shark brain [73]. The transporter(s) that are responsible for glucose secretion to the CSF by CP epithelial cells in mammals remains to be clarified.

Several studies demonstrated that the CP undergoes age-associated morphological, and functional alterations. CP epithelial cells exhibit reduced height, total volume, and length of apical villi, resulting in a more flattened appearance [74,75,76,77]. The accumulation of lipofuscin and thickening of the basement membrane, CP stroma, and vessel walls were also reported [74,75,76]. Reduced CSF production in the elderly has also been observed in humans, rats, and sheep [74,75,76,78]. The metabolic activity of the CP epithelial cells declines with aging, with CP epithelial cells isolated from old (24 months of age) rats showing decreased intracellular dehydrogenase activity as compared with younger (three to four months of age) rats [79]. Moreover, CP epithelial cells lacking cytochrome C oxidase then accumulate with age [80], whereas the expression of lactate dehydrogenase (necessary for anaerobic respiration) and succinate dehydrogenase (necessary for oxidative respiration) decrease [81,82], overall resulting in reduced energetic output in senescent CP epithelial cells [74,82]. These age-associated CP alterations are more prominent in AD, which are believed to be linked to the increasing Aβ burden in the CP [74,77,78,83,84]. By dynamic ^18^F-fluorodeoxyglucose (FDG) PET scans, AD subjects were found to have reduced ^18^F-FDG metabolism in the CP compared with subjects with amnestic mild cognitive impairment or healthy subjects [85]. This glucose metabolism imbalance in patients with AD may be attributed to a reduced glucose transport by GLUT1 [85]. CP epithelial cells need high energy to exert their homeostatic and secretory functions; therefore, they are equipped with a large number of mitochondria to meet their high energy demands [14,74]. A failure in glucose transport and subsequent metabolic derangement should affect various functions of CP epithelial cells, including CSF production, transport across the BCSFB, and the secretion of growth factors into the CSF [74]. Thus, impairments in glucose metabolism in the CP epithelial cells may contribute to age-related brain function decline, as well as to the pathophysiology of neurodegenerative diseases.

The presence of SGLT2 in CP epithelial cells raises the possibility that SGLT2 inhibitors, a new class of antidiabetic drugs, may have additional effects on the CNS [55]. SGLT2 inhibitors may modulate CSF production, the composition of glucose and/or sodium in the CSF, and/or glucose metabolism in the CP epithelial cells. Although no significant CNS side effects have been described for SGLT2 inhibitors, the impact of such compounds on the CP epithelial cells merits further investigation to ensure the better care and safety of T2DM patients. Studies using Sglt2 knockout and mutant mice (SAMP10-ΔSglt2) [86,87]—in particular, models of targeted knockdown in the CP epithelial and ependymal cells [88], could pave the way for a better understanding of the Sglt2 functions in these cells.

## 3. Fructose Transporters in Choroid Plexus Epithelial Cells

Fructose is an isomer of glucose, meaning that it shares the same molecular formula (C_6_H_12_O_6_), whereas it has a different stereochemical structure. Fructose most frequently holds an active open-chain structure and interacts more rapidly with proteins than glucose to form reactive adducts (Maillard reaction) [89,90,91]. Due to its enhanced reactivity, an excessive fructose intake can promote the formation of advanced glycation end products, resulting in diabetic complications and neurodegeneration [89,90,92]. Fructose consumption in Western diets has increased, mostly due to the intake of refined or processed sugars, such as high-fructose corn syrup [89,90,93]. Increased fructose ingestion correlates with the development of metabolic diseases such as obesity, insulin resistance, T2DM, nonalcoholic fatty liver disease, hypertension, hyperuricemia, and dyslipidemia [90,93,94]. Furthermore, results from animal studies suggest that high fructose intake may be linked to cognitive decline derived from insulin resistance, reduced neurogenesis, mitochondrial dysfunction, and a reduction of hippocampal plasticity [89,90,93,95,96,97,98]. Epidemiological studies have also suggested an association between the consumption of a Western diet (high in saturated fats and added sugars) with cognitive dysfunction in humans [99,100]. However, the short- or long-term effects of fructose intake on cognitive function in humans remain to be investigated [89,101].

Although researchers have been trying to better understand the fructose transport process from the blood to the brain, the collected data are still inconclusive. To date, GLUT5, which is the only fructose-specific facilitative glucose transporter [102], has been identified in the human BBB [103]. However, radiolabeled fructose injection into the rat common carotid artery resulted in no measurable uptake of fructose in the brain in the short period, suggesting the insignificant transport of fructose across the BBB [104].

There are seven SLC2A and two SLC5A family proteins that can transport fructose—specifically, GLUT2, GLUT5, GLUT7, GLUT8, GLUT9a/b, GLUT11, GLUT12, SGLT4, and SGLT5. Among these, five proteins/mRNAs have been identified in CP epithelial cells (GLUT5, GLUT8, GLUT9a/b, GLUT12, and SGLT5). In this section, we discuss the role of GLUT5, GLUT8, and SGLT5. GLUT9 will be discussed in the following section, as it has been recently recognized as a urate transporter. GLUT12 was already mentioned in the previous section.

### 3.1. GLUT5 (SLC2A5)

GLUT5 is a class 2 GLUT protein that was first identified in and initially cloned from the human small intestine [20,105]. Among nine members of the SLC2A and SLC5A family proteins able to transport fructose, GLUT5 is the sole fructose transporter, being unable to transport other sugars such as glucose, galactose, or mannose [102]. SLC2A5 mRNA and/or protein was identified in the small intestine, kidneys, fat, skeletal muscle, brain, and sperm [20,102].

In the brain, GLUT5/Glut5 is expressed in microglia [106], cerebellar Purkinje cells [107,108], the human BBB [103], rat hippocampus [95,109], rat cerebral cortex [107,108], rat olfactory bulb [108], rat optic tract and its terminal fields [107], and in rat vestibular and cochlear nuclei [107]. GLUT5 is the sole hexose transporter found in microglia [110]. An in-situ hybridization study further revealed that neurons expressing *Glut5* (cerebellar Purkinje cells; cerebral cortical neurons; and neurons in the granule cell layer, mitral cell layer, and periglomerular cells in the olfactory bulb) also expressed *Khk* and *Aldoc*, which encode enzymes that can catabolize fructose via the fructose-1-phosphate pathway [108], indicating the possibility that these neurons may use fructose, in addition to glucose, for energy production. However, it is still unclear whether high-fructose diets can induce Glut5 upregulation in the brain [95,102,109].

GLUT5/Glut5 has also been identified in ependymal cells [107,111] and CP epithelial cells [24,25,29,111]. A transcriptomic analysis of embryonic and adult rat CP revealed that *Slc2a5* was more expressed in adult CP [25,29]. Although a proteomic analysis revealed GLUT5/Glut5 expression in humans but not in rat CPs [24], our study revealed the presence of GLUT5/Glut5 protein in humans as well as in the rat CP and confirmed the expression of *Slc2a5* and the Glut5 protein in the mouse brain [111]. Moreover, immunohistochemical data demonstrated that GLUT5 is present in the apical side of the CP epithelial and ependymal cells (Figure 3a) [111].

### 3.2. GLUT8 (SLC2A8)

GLUT8 belongs to the class 3 GLUT proteins and is predominantly expressed in the testes and brain [20,113]. It has a high affinity for glucose, and fructose and galactose compete for its transport activity [20]. A recent study identified GLUT8 as a mammalian trehalose transporter [114,115]. GLUT8 can be found in intracellular compartments such as endosomes, lysosomes, and the endoplasmic reticulum, and no conventional signals, including stimulation by insulin, membrane depolarization, or activation of protein kinase A or C, can induce its translocation to the plasma membrane [113]. Thus, GLUT8 seems to be responsible for transporting sugars into or out of intracellular organelles [113].

GLUT8 has been detected in several brain regions, including the cerebral cortex, cerebellum, certain hypothalamic nuclei, the amygdala, dentate gyrus of the hippocampus, and subventricular areas [112,113,115]. GLUT8-expressing cells are mainly neurons [116], although astrocytic and/or microglial expression was also suggested [112,115]. *Slc2a8* knockout mice showed hyperactive behavior and increased arousal [117], as well as increased cell proliferation in the dentate gyrus without memory disturbances [118]. In addition to neuronal and glial cells, GLUT8/Glut8 was also found to be expressed by CP epithelial and ependymal cells in human and mouse brains [28]. The subcellular localization of GLUT8 in these cells was also cytoplasmic (Figure 3b). A transcriptomic analysis detected a higher expression of *Slc2a8* in the adult rat CP than in the embryonic CP [29].

### 3.3. SGLT5 (SLC5A10)

SGLT5 is expressed almost exclusively in the kidney cortex and has a relatively high affinity and capacity for mannose and fructose relative to glucose and galactose [42,43,119]. SGLT5, along with SGLT4, is present in the luminal membrane of the proximal tubule and contributes to the reabsorption of fructose [120]. Transcriptome studies on the rat CP showed that the *Slc5a10* transcript was present at low levels only in the embryonic stage [25,29]. The physiological significance of SGLT5 in the CP remains unknown.

### 3.4. Physiological and Pathophysiological Considerations of Fructose Transporters in Choroid Plexus Epithelial Cells

Estimates of blood fructose concentrations can vary significantly [102], with serum fructose concentrations in healthy, diabetic, and nondiabetic subjects being reported of 8.1 ± 1.0 μM, 12.0 ± 3.8 μM, and 7.7 ± 1.6 μM, respectively [121]. Moreover, serum fructose levels in heathy humans consuming high-fructose or high-sucrose diets can reach 200–500 μM, which is still an order of magnitude lower than the normal blood glucose level (5.5 mM) [102]. These low blood fructose levels can be attributed to the lower intestinal absorption rate compared with glucose and to the efficient clearance of fructose, mainly by the liver (50–70%) and, to a lesser extent, by the kidneys (20%) [102]. Interestingly, fructose concentrations in the CSF are on the order of 100 μM, exceeding the plasma fructose level [122,123]. In addition, a close correlation between glucose and fructose concentrations in CSF was reported [123]. These observations may be explained by the recent report that fructose is produced in the human brain via the polyol pathway [124].

The physiological functions of fructose in the brain are not fully understood. In Drosophila, nutritional carbohydrate ingestion results in increased hemolymph fructose levels, which, in turn, act on a nutrient sensor [125]. Whether fructose plays a similar role in nutrient sensing in mammalian brains warrants investigation. However, the intraventricular injection of fructose showed a stimulation of the food intake in rats, whereas glucose delivered directly to the brain had an opposite effect [97,101,126]. These studies suggest a direct action of fructose on the hypothalamic feeding center. Furthermore, fructose may represent an alternative energy source by neuronal or glial cells [102,108,112,127] and may alter the neuronal and glial interactions [124].

GLUT5 is expressed in the apical side of the CP epithelial and ependymal cells (Figure 3a) [111], whereas no fructose transporters were described in the basolateral membrane of the CP epithelial cells. Considering the higher fructose concentration in CSF than in serum, it can be speculated that GLUT5 in the CP epithelial cells can import fructose from the CSF to CP epithelial cells rather than transport it from the blood to the CSF. Adult rat and human CP transcriptomic analyses revealed low expressions of *Khk* and *Aldoc* [29,128], which are genes involved in fructose catabolism via the fructose-1-phosphate pathway [108], suggesting that CP epithelial cells may use fructose for energy production.

Mice and rats exposed to high-fructose feeding exhibited reduced hippocampal neurogenesis and synaptic plasticity, concomitantly with increased circulating tumor necrosis factor-α levels, and oxidative damage in the hippocampus [96,97]. Due to their close proximity to the CP and direct exposure to the CSF, the subventricular and hippocampal subgranular zones, the two major sites where adult neurogenesis occurs, are under the influence of the CSF contents [6,75]. Given its high reactivity, CSF fructose may cause macromolecular modifications in neural progenitor cells residing in these adult neurogenic niches [89,90,91]. Provided that fructose is transported from the CSF to CP epithelial cells, a decrease in GLUT5 expression in CP epithelial cells could cause an increase in the CSF fructose concentration, which, in turn, could result in the impairment of periventricular structures, including the hippocampus. Therefore, changes in the GLUT5 expression during aging and in neurodegenerative diseases, as well as the direction of fructose transport across the BCSFB, deserve further investigation.

## 4. Urate Transporters in Choroid Plexus Epithelial Cells

Uric acid (C_5_H_4_N_4_O_3_) is the final product of purine (adenosine and guanine) metabolism in humans and some higher primates [129,130]. Uric acid is endogenously derived from nucleic acids and can be exogenously obtained from animal proteins and fructose catabolism [108,129,131,132]. Not only is hyperuricemia the main determinant of gout, it is also implicated in the pathogenesis of metabolic syndrome, hypertension, atherosclerosis, coronary artery disease, heart failure, and atrial fibrillation [129,131]. Hyperuricemia is currently considered to contribute to the cardiovascular risk in patients with hypertension [133], despite some evidence suggesting that urate per se is not a risk factor but only a marker of cardiovascular damage [129]. Regardless of its pathogenic potential, 90% of the urate filtered in the kidney glomeruli is reabsorbed, suggesting that urate may also have a physiological role [131,134]. Urate is the most abundant aqueous antioxidant, accounting for up to 60% of the plasma antioxidative capacity [131,134] and has gained interest due to its neuroprotective properties [17]. Urate levels are inversely associated with the risk of Parkinson’s disease (PD) and AD, and with the progression of PD, amyotrophic lateral sclerosis (ALS), Huntington’s disease, and multiple system atrophy [17,135,136]. A recent meta-analysis revealed low serum urate levels in patients with multiple sclerosis and neuromyelitis optica [137]. Owing to its neuroprotective potential, therapeutic applications of urate have been investigated for PD, ALS, and acute ischemic stroke [17,18,138]. In contrast, some studies suggested that hyperuricemia may be an independent risk factor for white matter atrophy and memory deficits [139,140] by acting as a potent inflammatory stimulant, thereby resulting in neuroinflammation and hippocampal gliosis [141].

Urate needs to pass through the BBB and/or BCSFB by specific transporters to enter the CSF and brain ISF. Among urate transporters, GLUT9/SLC2A9, urate transporter 1 (URAT1/SLC22A12), and breast cancer resistance protein (BCRP/ABCG2) were reported to play important roles in the regulation of serum uric acid [130,134] and are all expressed in the BBB and/or BCSFB [142,143,144].

### 4.1. GLUT9 (SLC2A9)

GLUT9 is a class 2 transporter that was initially considered a glucose and fructose transporter [20]; however, its substrate is now known to be urate instead [20,134,145,146]. Two distinct N-terminal isoforms, GLUT9a (540 amino acids) and GLUT9b (511 amino acids), are generated by alternative splicing of the 5′ ends of *SLC2A9*. These two variants are differently expressed, with GLUT9a being ubiquitously expressed and present in the basolateral membrane of the proximal tubular epithelium of the human kidneys, whereas GLUT9b is expressed only in the liver and kidneys, specifically in the apical membrane of the collecting duct [20,134,147]. Genome-wide association studies to find gene loci associated with urate levels identified SLC2A9 as the major locus that was significantly associated with gout [20,131,134,145,148]. Inactivating mutations in *SLC2A9* cause hypouricemia with a massive renal excretion of urate in humans [20,146], suggesting that GLUT9 is essential for urate reabsorption.

An in-situ hybridization study of adult mouse brains revealed that *Slc2a9* is expressed by Purkinje cells, neurons in the dentate gyrus, hippocampal pyramidal neurons, cells in the cerebral cortices, and neurons in the olfactory bulb [108]. Glut9 was also detected by Western blotting of membrane fractions of tissue samples where *Slc2a9* was detected [108]. Glut9 was also found to be present in the mouse substantia nigra and MES23.5 cell line, a hybrid of rat embryonic mesencephalon cells with mouse neuroblastoma cells [149].

An immunohistochemical analysis of mouse brain frozen sections revealed the expression pattern of Glut9 in ependymal cells, tanycytes lining the floor of the ventral third ventricle, and NeuN-positive cells in the brain parenchyma, although submembrane localization (apical or basolateral) was inconclusive [143]. Furthermore, immunohistochemistry using a methacarn-fixed mouse brain revealed Glut9 expression on the luminal membrane of the brain capillary endothelium [143]. Although these approaches failed to show Glut9 expression in the CP, the presence of *Slc2a9* was identified not only in ependymal cells but, also, in CP epithelial cells by highly sensitive in-situ hybridization [143]. We have recently reported the expression of GLUT9 in CP epithelial and ependymal cells of autopsied human brains; however, immunoreactivity signals in the CP were weak and heterogeneous among subjects [142]. In our data, GLUT9 was mainly localized on the apical side of CP epithelial and ependymal cells (Figure 4a) [113].

Recent studies suggest a uricosuric effect of SGLT2 inhibitors [130,150,151], although SGLT2 does not appear to have the ability to transport urate [152]. It is believed that a high glucose concentration in the lumen of proximal tubules may suppress the activity of GLUT9b [151,152]. Nevertheless, previous studies indicated that excessive glucose or fructose would not compete against urate for GLUT9 transport [20,146].

### 4.2. URAT1 (SLC22A12)

URAT1 is a member of the SLC22A transporter family, which has a pivotal role in the movement of small molecules (endogenous metabolites, drugs, and toxins) between tissues and interfacing fluids in the renal proximal tubules, hepatocytes, and CP [153]. URAT1 mediates urate uptake in exchange for intracellular organic anions such as lactate and nicotinate and is a main mediator of urate reabsorption in the kidneys [154]. URAT1 is located in the apical membrane of the proximal tubular epithelial cells and is a drug target for hyperuricemia [154]. Moreover, mutations in *SLC22A12* are causative of idiopathic renal hypouricemia [154]. A human transcriptomic analysis revealed that *SLC22A12* is mainly expressed in the kidneys and is present in low levels in adipose tissue, the colon, liver, stomach, brain, and testes [155].

In the mouse brain, Urat1 expression was mainly detected in the apical surfaces, including the cilia of ependymal cells lining the ventricles and aqueduct, except for tanycytes in the ventral third ventricle [144]. The weak immunoreactivity in the CP and brain parenchyma was considered nonspecific, as the detected signal was similarly observed in *Slc22a12* knockout mice [144]. Although mouse Urat1 was demonstrated to be present in the brain capillaries and CP by immunoblotting [156], immunohistochemistry data failed to confirm Urat1 expression in these structures [144]. In contrast, in-situ hybridization revealed *Slc22a12* signals in the CP and brain parenchyma, as well as in ependymal cells [143]. Our immunohistochemical study using autopsied human brains further revealed that URAT1 is located in the basolateral side of CP epithelial cells (Figure 4b) [142].

### 4.3. BCRP (ABCG2)

BCRP/ABCG2 is a member of a large ATP-binding cassette (ABC) family, which are categorized as active transporters that use ATP to transport substrates across the membrane [4]. BCRP/ABCG2 is expressed in the placenta, liver, intestines, kidneys, heart, and brain and is involved in the cellular efflux of a large number of chemically and structurally diverse compounds, including sulfoconjugated organic anions, hydrophobic and amphiphilic drugs, and Aβ peptides [4,130,157]. Furthermore, BCRP/ABCG2 plays an important role in the renal and gastrointestinal excretion of urate, whereas the dysfunction of BCRP/ABCG2 has proven to be a major cause of hyperuricemia and gout [130,134].

In the brain, BCRP/ABCG2 was identified at both the BBB and BCSFB [4]. It is expressed on the luminal membrane of endothelial cells in humans, monkeys, mice, and rats [4,11,143,158,159,160]. Quantitative proteomic analyses revealed that the levels of BCRP/Bcrp at the BBB are highest in monkeys (14–16 fmol/μg protein) [158,161], followed by humans (3.8–10.9 fmol/μg protein) [162] and rats and mice (4–5 fmol/μg protein) [158,161]. An age-dependent increase in Bcrp expression in the brain capillaries was observed in cynomolgus monkeys [161]. BCRP/Bcrp was found in the CP of humans, nonhuman primates, mice, and rats. Transcriptomic analyses of mouse and rat CP revealed that *Abcg2* was significantly upregulated in the embryonic CP as compared with the adult CP [29,56,163]. An in-situ hybridization study confirmed the expression of *Abcg2* in the mouse CP but not in ependymal cells [143]. A quantitative proteomic analysis demonstrated that BCRP/Bcrp is present in the human and rat CP, with two-fold higher expression in humans (0.7 fmol/μg protein) than in rats (0.3 fmol/μg protein) [24]. BCRP levels in the CP, however, were significantly lower than those in the BBB [4,163]. Immunohistochemical studies indicated that BCRP/Bcrp is located in the CSF side membrane in human and mouse CP epithelial cells (Figure 4c) [143,159,163,164]. The apical localization of ABCG2 implies that it would transport its substrates, including urate and Aβ peptides, into the CSF [143,163].

### 4.4. Physiological and Pathophysiological Considerations of Urate Transporters in Choroid Plexus Epithelial Cells

The normal serum reference levels of uric acid are 1.5–6.0 mg/dL (89–357 μM) in women and 2.5–7.0 mg/dL (149–417 μM) in men [134], whereas the CSF urate concentration is 10- to 20-fold lower than in plasma; however, a positive correlation between plasma and CSF urate levels was reported [165,166]. Urate concentrations in rat CSF (6.13 ± 0.68 μM) [167] were found to be higher than in the rat brain ISF (approximately 1 μM) [168]. Moreover, rats fed a high uric acid diet exhibited an increase in urate levels in the hippocampus as compared to those fed a control diet [141], suggesting the presence of a transporting system of dietary uric acid into the brain. Taking into consideration the herein described localization of urate transporters at the BBB and BCSFB, it is plausible that brain ISF urate may be mainly derived from the CSF [142,143,144]. BCRP/ABCG2 is the main urate transporter at the BBB, which is expressed on the luminal membrane of brain capillary endothelial cells and likely acts to excrete urate to the blood, rather than to transport it into the brain parenchyma [4,143,158]. In contrast, immunohistochemical and in-situ hybridization studies of human and mouse brains revealed that URAT1/Urat1, GLUT9/Glut9, and BCRP/Bcrp are all expressed in CP epithelial and/or ependymal cells [142,143,144]. Although there are some differences in the protein expression pattern between humans and mice, urate seems to be transported from the blood to the CSF via CP epithelial cells (see Figure 5) and then from the CSF to the brain parenchyma via ependymal cells [142,143,144]. Considering the inverse association between urate levels and the risk and progression of neurodegenerative diseases, including AD and PD, and the antioxidative and neuroprotective properties of urate, it is intriguing to investigate whether urate transporters in these cells exhibit expression changes with aging or in patients with age-associated neurodegenerative disorders.

## 5. Hexose and Urate Transporters in the Choroid Plexus Epithelium: Comparison with the Renal Proximal Tubular Epithelium

Due to their morphological and functional similarities, CP is impressively described as the “kidney” of the brain [10]. CP epithelial cells share morphological features with the epithelia involved in extensive transcellular transport, such as the proximal tubular epithelial cells, including the presence of brush borders or microvilli on the apical membrane, basolateral infoldings (basal labyrinth), the formation of tight junctions between neighboring cells, and the abundance in mitochondria [1,5,6,10,169,170]. They both form a leaky barrier with transepithelial electrical resistance in the intermediate range due to the expression of claudin 2, the only pore-forming claudin [1,5,6], and they are responsible for purifying the extracellular fluid (plasma for the proximal tubular epithelium and CSF for the CP epithelium), resulting in the transfer of a relatively large volume of fluid. One of the prominent features of CP epithelial cells is their apparent inverse polarization: some of the “basolateral transporters”, such as Na^+^,K^+^-ATPase and NHE1, are expressed in the apical membrane of CP epithelial cells (Figure 5) [10,13]. However, not all “basolateral transporters” show inverted localization, which is the case of AE2, KCC3, and NBCn1, which occupy the basolateral membrane domain in both the CP and proximal tubular epithelium (Figure 5) [13]. A recent study investigating the subcellular distribution of an array of proteins known to be apical and basolateral determinants further revealed that CP epithelial cells are normally polarized according with the localization of the major polarity protein complexes and the membrane phospholipids phosphatidylinositol 4,5-bisphosphate and phosphatidylinositol 3,4,5-triphosphate [5,171]. However, several deviations from other polarized epithelia, including (1) the expression of P- and N-cadherins instead of E-cadherin, (2) the apical localization of Lgl2, one of the proteins that establish and maintain the basolateral membrane domain, (3) the lack of AP-1B expression, which is necessary for basolateral membrane recycling, and (4) the apical localization of syntaxin-4, a SNARE protein providing specificity to the fusion of transport vesicles to the basolateral membrane, were identified in the CP epithelial cells [5,171]. Further studies are warranted to clarify which molecular deviations are determinant for the atypical submembrane localization of certain transporters in CP epithelial cells.

The expression pattern of hexose and urate transporters in CP epithelial cells resembles that in proximal tubular epithelial cells, although some transporters have different submembrane localizations. Our immunohistochemical study showed that GLUT9 and URAT1 exhibited an inverted localization between CP epithelial cells and proximal tubular epithelial cells (Figure 5) [142]. SGLT2 may also be present in the basolateral membrane in CP epithelial cells (Figure 2), being another example of inverted polarity of the CP epithelial cells (Figure 5). In contrast, GLUT5 [102] and BCRP [172] were localized on the apical side in both cell types (Figure 5). These partial deviations in the submembrane localization of urate and glucose transporters in CP epithelial cells may originate from the expression differences of cell polarity-determinant proteins, as described above [5,171].

## 6. Conclusions and Future Perspectives

In summary, CP epithelial cells express (1) GLUT1 and SGLT2 as glucose transporters, mainly on the basolateral membrane, (2) fructose transporter GLUT5 on the apical membrane, (3) fructose and glucose transporter GLUT8 in the cytoplasmic membrane compartment, and (4) urate transporters GLUT9 and BCRP/ABCG2 on the apical membrane and URAT1 on the basolateral membrane (Figure 5). The transporters and ion channels expressed in CP epithelial cells that are involved in the secretion of CSF have been extensively studied [5,10,13,23]. Recent knowledge on the advances on glucose, fructose, and urate transporters expressed in CP epithelial cells highlight their potential relevance for the physiology and pathophysiology of the CNS. Despite of some differences in submembrane localization, the expression of common transporters for hexose and urate in CP epithelial cells and renal proximal tubular epithelial cells provides additional evidence for the similarity of these epithelia. However, the physiological and pathophysiological aspects of these transporters expressed in CP epithelial cells are still largely unknown and await further investigation. How and to which direction CP epithelial cells transport glucose, fructose, and urate across the BCFSB through these transporters should be clarified in future studies. Although it is difficult to investigate the transporter function in CP epithelial cells in humans, the analyses of the CSF composition and glucose metabolism in subjects who have mutations in these transporter genes or those who are treated with specific inhibitors (for example, those having T2DM and treated with SGLT2 inhibitors) may clarify some functional roles of these transporters. The functional decline in these transporters in CP epithelial cells could impair the homeostasis of the brain microenvironment and affect the periventricular structures, including the hippocampus, thereby contributing to the pathogenesis of cognitive impairment and dementia. Although CP may only have an indirect influence on the pathogenesis of these conditions, a better understanding of how the functional impairment of the CP epithelium can modulate pathological processes in the brain could reveal new potential therapeutic targets in the CP-CSF system for treating and preventing dementia and other age-associated neurodegenerative disorders [13,74,75,76,78].

## Figures and Tables

**Figure 1 ijms-21-07230-f001:**
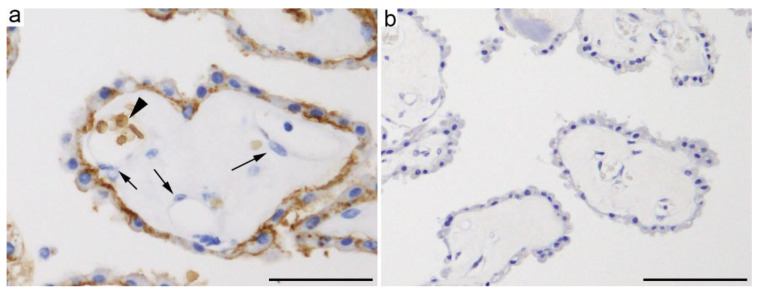
Representative microphotographs of immunohistochemical staining for (**a**) GLUT1 (clone SP168, ab150299, Abcam, Cambridge, UK, 1:200, antigen retrieval with citrate buffer (pH 6)) and (**b**) GLUT12 (bs-2540R, Bioss, Boston, MA, USA) [28] in human choroid plexus (CP). (**a**) GLUT1 is mainly located in the basolateral membrane of CP epithelial cells, whereas no GLUT1 is detected in capillary endothelial cells of the CP parenchyma (arrows). Erythrocytes in the capillary show high GLUT1 expression (arrowhead). (**b**) No apparent GLUT12 immunoreactivity is observed in CP epithelial cells. Scale bars: 50 μm.

**Figure 2 ijms-21-07230-f002:**
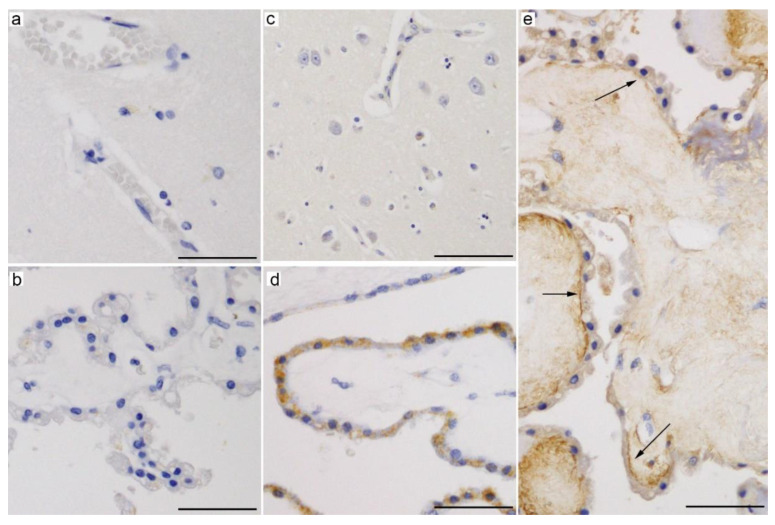
Representative microphotographs of immunohistochemical staining for (**a**,**b**) SGLT1 (ab14685, Abcam) [55], (**c**,**d**) SGLT2 (NBP1-92384, Novus Biologicals, Centennial, CO, USA) [55], and (**e**) SGLT2 (ab85626, Abcam, 5 μg/mL, antigen retrieval with citrate buffer (pH 6)) in autopsied human brains. Endothelial cells of brain microcapillaries express neither SGLT1 (**a**) nor SGLT2 (**c**). CP epithelial cells are devoid of SGLT1 immunoreactivity (**b**). In contrast, SGLT2 is expressed in CP epithelial cells, with an intracytoplasmic granular pattern (**d**), or along the plasma membrane of the basal side (**e**, arrows), depending on the antibodies used. Scale bars: 50 μm.

**Figure 3 ijms-21-07230-f003:**
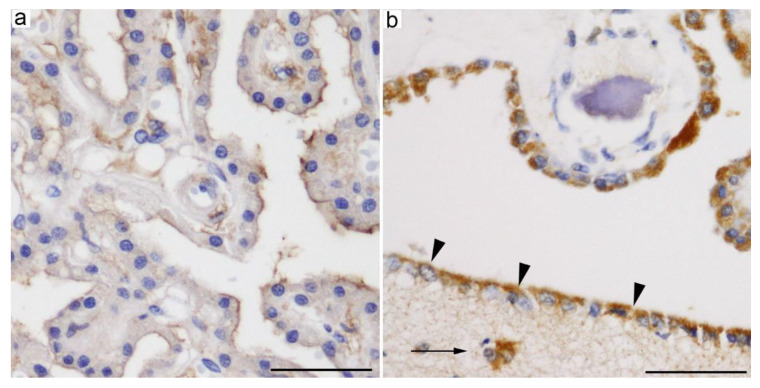
Representative microphotographs of immunohistochemical staining for (**a**) GLUT5 (18905, IBL, Takasaki, Japan) [111] and (**b**) GLUT8 (bs-4241R, Bioss) [28] in human CP. (**a**) GLUT5 is mainly located in the apical side of CP epithelial cells. (**b**) Intracytoplasmic granular immunoreactivity for GLUT8 is observed in CP epithelial cells, as well as in ependymal cells (arrowheads) and subependymal glial cells (arrow) [112]. Scale bars: 50 μm.

**Figure 4 ijms-21-07230-f004:**
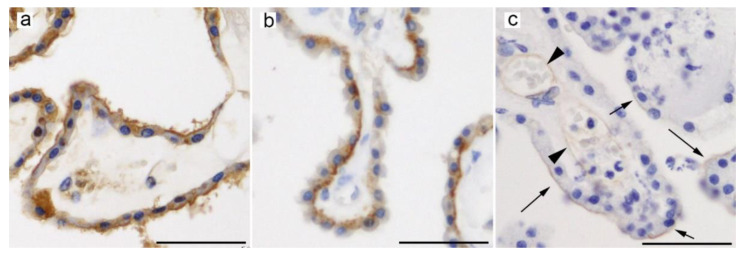
Representative microphotographs of immunohistochemical staining for (**a**) GLUT9 (ab104623, Abcam) [142], (**b**) URAT1 (BMP064, MBL, Nagoya, Japan) [142], and (**c**) BCRP/ABCG2 (clone BXP-21, ab3380, Abcam, 1:100, antigen retrieval with citrate buffer (pH 6)) in the human CP. (**a**) GLUT9 is mainly located in the apical side of CP epithelial cells. (**b**) Immunoreactivity for URAT1 is observed in the basolateral side of CP epithelial cells. (**c**) Along with endothelial cells in the CP parenchyma (arrowheads), BCRP/ABCG2 is expressed on the apical surface of CP epithelial cells (arrows). Scale bars: 50 μm.

**Figure 5 ijms-21-07230-f005:**
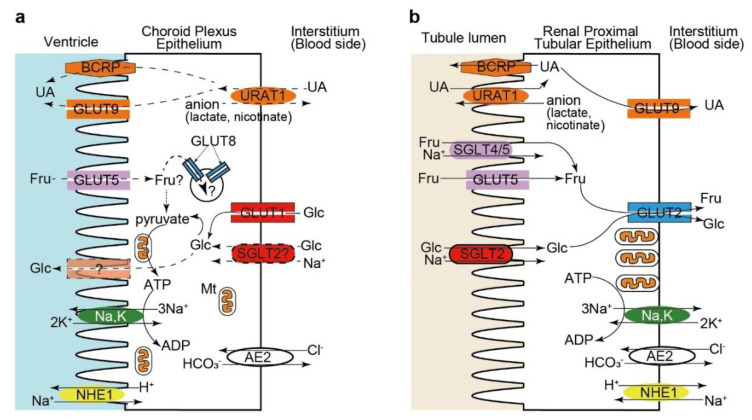
Polarized distribution of glucose (Glc), fructose (Fru), and urate (UA) transporters in choroid plexus (CP) epithelial cells (**a**) and renal proximal tubular epithelial cells (**b**). Putative directions of Glc, Fru, and UA transport through CP epithelial cells are indicated by dashed arrows. The transporter for Glc in the apical side of CP epithelial cells in mammals is still uncertain (indicated by a semitransparent rectangle labeled with a question mark). GLUT8 is expressed in the intracellular membranes, such as lysosomes, endoplasmic reticulum, and endosomes. As examples for the partially inverted distribution of transporters between CP epithelial cells and renal proximal tubular epithelial cells, the localization of Na^+^,K^+^-ATPase (Na and K) and NHE1 is indicated. AE2 is located in the basolateral membrane in both cells. Mt, mitochondria.

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
