# Peer review of "Glucose, Fructose, and Urate Transporters in the Choroid Plexus Epithelium"

_ijms, 2020, doi:10.3390/ijms21197230_

Round 1
Reviewer 1 Report
In the current manuscript (No. ijms-927781), the authors make a review on the glucose, fructose, and urate transporters of the choroid plexus epithelial cells and relate their expression and function to processes of brain aging and neurodegeneration.
In general this is a good and timely manuscript covering relevant literature on a very interesting topic, such as the involvement of the CP in brain homeostasis and disease. Still, there are some concerns that if addressed could improve the clarity of the manuscript.
- At the beginning of the introduction (line 42) may be beneficial to shortly mention the specific macromolecules that cross through the BCSFB and BBB and if some sort of barrier specificity exist.
- Also, specificity is needed in line 48: “plasma components, blood cells, and pathogens”.
- More clarity is needed within the introduction. For example, in line 57 and 58, the authors comment on the equipment of endothelial and CP cells, then, the next line starts with a negation of that to compare BBB with BCSFB, which is confusing.
- Image sources should be given in figure legends. This is more relevant if they are not original work (i.e., published elsewhere and not made for the review) from the authors.
- Even when the reviews report previous work in which some involvement is suggested. No conclusive results exist. Moreover, most of the research conducted to directly relate CP molecular function has been conducted in animals or animal models, or even in isolated tissue samples, which, unfortunately, given their biological, evolutive differences with humans still makes such assumptions hard to argument.
- Some recent work (See Gonzalez-Escamilla et al., 2020; Walker et al., 2018; Guillen et al., 2020) in which has been proposed that FDG-PET findings are very unspecific of AD-related cognitive decline, but rather related to general dementia. Even when the sole in of the CP in other brain pathologies remains unclear, the authors should not be focus on only AD.
- It is unclear the level at which all these molecules are able to cross the BCSFB is unclear. In so, translation from organ production/expression to brain physiological and pathophysiological conditions is still challenging.
- A part with current limitations and perspectives arising from the review is completely missing.
References:
- Gonzalez-Escamilla G, Miederer I, Grothe MJ, Schreckenberger M, Muthuraman M, Groppa S; Alzheimer’s Disease Neuroimaging Initiative. Metabolic and amyloid PET network reorganization in Alzheimer's disease: differential patterns and partial volume effects. Brain Imaging Behav. 2020;3. doi: 10.1007/s11682-019-00247-9.
- Guillén EF, Rosales JJ, Lisei D, Grisanti F, Riveron M, Arbizu J. Current role of 18F-FDG-PET in the differential diagnosis of the main forms of dementia. Clin Transl Imaging 2020;8, 127–140. https://doi.org/10.1007/s40336-020-00366-0
- Walker Z, Gandolfo F, Orini S, Garibotto V, Agosta F, Arbizu J, Bouwman F, Drzezga A, Nestor P, Boccardi M, Altomare D, Festari C, Nobili F; EANM-EAN Task Force for the recommendation of FDG PET for Dementing Neurodegenerative Disorders. Clinical utility of FDG PET in Parkinson's disease and atypical parkinsonism associated with dementia. Eur J Nucl Med Mol Imaging. 2018;45(9):1534-1545. doi: 10.1007/s00259-018-4031-2
Author Response
Response to Reviewer 1
We appreciate your constructive comments and suggestions which have significantly helped us improve our manuscript. As indicated in our responses below, we have taken all your comments and suggestions into consideration when revising our manuscript.
- At the beginning of the introduction (line 42) may be beneficial to shortly mention the specific macromolecules that cross through the BCSFB and BBB and if some sort of barrier specificity exist.
Response: Thank you for your comment. In this sentence, we describe on the passage of the molecules through the fenestrated capillaries in the choroid plexus stroma. We changed the phrase “allowing the free passage of macromolecules” to “allowing a facilitated diffusion of molecules with molecular weight up to ~800 kDa” (p1, lines 42-43) with new reference citation.
- Also, specificity is needed in line 48: “plasma components, blood cells, and pathogens”.
Response: Thank you for your comment. According to your suggestion, we revised the phrase “plasma components, blood cells, and pathogens” to “plasma proteins such as albumin and circulating blood cells (erythrocytes and leukocytes)”. (p2, lines 48-49)
- More clarity is needed within the introduction. For example, in line 57 and 58, the authors comment on the equipment of endothelial and CP cells, then, the next line starts with a negation of that to compare BBB with BCSFB, which is confusing.
Response: Thank you for your comment. According to your suggestion, we revised the phrase “the BBB and BCSFB” to “these two cells” (p2, line 59).
- Image sources should be given in figure legends. This is more relevant if they are not original work (i.e., published elsewhere and not made for the review) from the authors.
Response: Thank you for your comment. We took all microphotographs presented in Figures 1-4 originally from specimens prepared for this manuscript with the published staining conditions. Although immunolocalization of GLUT1 in the choroid plexus (Figure 1a) is a classical finding and not our original one, we reconfirmed in this manuscript the published basolateral localization in choroid plexus epithelial cells and the absence in the capillary endothelial cells using a currently available antibody. We added the references related to the presented microphotographs to the figure legends in Figures 1-4. Further, experimental conditions (antibody dilution and methods of antigen retrieval) were described briefly for GLUT1, SGLT2 (Abcam ab 85626), and BCRP/ABCG2, which have not been presented in our previous papers, in the figure legends.
- Even when the reviews report previous work in which some involvement is suggested. No conclusive results exist. Moreover, most of the research conducted to directly relate CP molecular function has been conducted in animals or animal models, or even in isolated tissue samples, which, unfortunately, given their biological, evolutive differences with humans still makes such assumptions hard to argument.
Response: Thank you for your comment. We presented Figure 5 and Concluding Remarks (Conclusions and Future Perspectives in a revised version) section as the conclusive summary of the manuscript. We further summarized at the beginning of the last section of the current experimental findings on the expression of the transporters for hexose and urate (p14, lines 565-569).
We understand your concern about the extrapolation of results of animal studies and those using isolated tissues to human biology. Investigation of CSF composition or glucose metabolism in humans with mutations in the transporters or having specific inhibitors (such as SGLT2 inhibitors for the treatment of DM) may help understand the functions of these transporters in human CP. These descriptions are added in the final section (p14, lines 578-582).
- Some recent work (See Gonzalez-Escamilla et al., 2020; Walker et al., 2018; Guillen et al., 2020) in which has been proposed that FDG-PET findings are very unspecific of AD-related cognitive decline, but rather related to general dementia. Even when the sole in of the CP in other brain pathologies remains unclear, the authors should not be focus on only AD.
Response: Thank you for your comment. We agree that FDG-PET findings are not necessarily specific to AD. Unfortunately, to our knowledge, there are few published papers on the neurodegenerative diseases-related changes in CP relating to the hexose and urate transport/metabolism, except a paper reporting a decrease in glucose metabolism in CP in AD patients (Daouk J et al Exp Gerontol 77:62-68, 2016). Further, morphological changes in CP epithelial cells have been mainly described in AD patients. This may be the reason why you felt that we focused only on AD in this review. As you suggested, morphological changes in CP epithelial cells and hexose/urate transporter expression could relate to other neurodegenerative and dementing diseases. Thus, to avoid misunderstanding, we deleted the phrase “, including Alzheimer’s disease” (p1, line 31) in the last part of Abstract and “Alzheimer’s disease” from the Keywords: instead, we added “neurodegenerative diseases” to the Keywords (p1, line 33).
- It is unclear the level at which all these molecules are able to cross the BCSFB is unclear. In so, translation from organ production/expression to brain physiological and pathophysiological conditions is still challenging.
Response: We agree with your comment. The presented molecular flows in Figure 5a are hypothetical ones estimated based on the characteristics of the transporters and concentration gradient between the membrane, thus depicted in dashed lines. The verification of these hypothesis is the future research tasks, as described in the final section (p14, lines 575-576).
- A part with current limitations and perspectives arising from the review is completely missing.
Response: Thank you for your comment. We described on the limitations and future perspectives in the last section (p14, lines 575-582).
Reviewer 2 Report
This is a good review about an interesting topic, such as the role of glucose, fructose and urate transporters in choroid plexus cells. The manuscript is well written, in an original and appropriate way. Noteworthy, this manuscript has the capacity to condensate a complicated topic in a reasonable length. I believe that this review is designated to be cited by several other scientists because it gives a complete overview about this topic.
Author Response
Response to Reviewer 2
We appreciate you for reading our review manuscript and your valuable comments.
Reviewer 3 Report
This is a very thorough review on the expression and function of glucose, fructose and urate transporters in the CP which gives the reader a very good introduction to the field.
A suggestion to the authors is to embed the specific discussions from the “physiological and pathophysiological considerations” of for example glucose into the part about localization of the transporters. And I suggest focusing the chapters on localization on only brain and a few tissues where relevant. The section about localization becomes a bit difficult to read because of all the information that is not really essential to the review. The strongest parts are the introduction and the physiological and pathophysiological considerations.
I am a bit confused about the figures in the review. Have these been published previously or are they only for the purpose of this review? If made for this review I need information on how the stainings were performed and how the specificity of the antibodies was assessed.
Author Response
Response to Reviewer 3
We appreciate your constructive comments and suggestions which have significantly helped us improve our manuscript. As indicated in our responses below, we have taken all your comments and suggestions into consideration when revising our manuscript.
A suggestion to the authors is to embed the specific discussions from the “physiological and pathophysiological considerations” of for example glucose into the part about localization of the transporters. And I suggest focusing the chapters on localization on only brain and a few tissues where relevant. The section about localization becomes a bit difficult to read because of all the information that is not really essential to the review. The strongest parts are the introduction and the physiological and pathophysiological considerations.
Response: Thank you for your suggestion. We put the sections of physiological and pathophysiological considerations on glucose, fructose and urate transporters as the subsections after the sections describing the localization of each transporter (i.e. physiological and pathophysiological considerations on glucose transporters as subsection 2.3., on fructose transporters as 3.4., and on urate transporters as 4.4.). We also revised the descriptions on localization of each transporters outside the CNS as follows.
- 2.1.2 GLUT12: p3, lines 117-118: …and is expressed in the heart, prostate, skeletal muscle, placenta, small intestine, adipose tissue, human and mouse brain at very low levels, and in the mouse CP →…and is expressed in human and mouse brain at very low levels and in the mouse CP, as well as in the heart, prostate, skeletal muscle, placenta, small intestine, and adipose tissue
- 2.1.4 GLUT4: GLUT4 dramatically changes its localization from intracellular membrane compartments to the plasma membrane upon insulin stimulation, with defects in GLUT4 translocation under insulin stimulation being known as peripheral insulin resistance [18,37]. → Deleted.
- 2.2.1 SGLT1: p4, lines 153-154: SGLT1 is expressed in the intestinal epithelial cells, where it absorbs dietary D-glucose and D-galactose from the gut lumen [17,42,44,45]. Further, SGLT1 is localized in the apical membrane of the S3 segment of the renal proximal tubule, where it ensures the absorption of all remaining glucose after the glomerular filtrate already passed the scrutiny of the SGLT2 (SLC5A2) in the S1/S2 segment [17,19,45–47]. →SGLT1 is expressed in the intestinal epithelial cells [19, 44, 46, 47] and in the apical membrane of the S3 segment of the renal proximal tubule [19, 21, 47-49].
I am a bit confused about the figures in the review. Have these been published previously or are they only for the purpose of this review? If made for this review I need information on how the stainings were performed and how the specificity of the antibodies was assessed.
Response: Thank you for your comment. We took all microphotographs presented in Figures 1-4 originally from specimens prepared for this manuscript with the published staining conditions. Although immunolocalization of GLUT1 in the choroid plexus (Figure 1a) is a classical finding and not our original one, we reconfirmed in this manuscript the published basolateral localization in choroid plexus epithelial cells and the absence in the capillary endothelial cells using a currently available antibody. We added the references related to the presented microphotographs to the figure legends in Figures 1-4. Further, experimental conditions (antibody dilution and methods of antigen retrieval) were described briefly for GLUT1, SGLT2 (Abcam ab 85626), and BCRP/ABCG2, which have not been presented in our previous papers, in the figure legends. The specificity of these antibodies has been assessed by the following papers.
GLUT1(clone SP168): Yu, L. et al. Core pluripotency factors promote glycolysis of human embryonic stem cells by activating GLUT1 enhancer. Protein Cell 10:668-680, 2019.
SGLT2 (Abcam 85626): Suga, T. et al. SGLT1 in pancreatic alpha cells regulates glucagon secretion in mice, possibly explaining the distinct effects of SGLT2 inhibitors on plasma glucagon levels. Mol Metab 19:1-12, 2019.
BCRP/ABCG2 (clone BXP-21): Maliepaard, M. et al. Subcellular localization and distribution of the Breast Cancer Resistance Protein transporter in normal human tissues. Cancer Res 61:3458-3464, 2001.